# Myocardial Deformation Analysis in *MYBPC3* and *MYH7* Related Sarcomeric Hypertrophic Cardiomyopathy—The Graz Hypertrophic Cardiomyopathy Registry

**DOI:** 10.3390/genes12101469

**Published:** 2021-09-23

**Authors:** Viktoria Höller, Heidelis Seebacher, David Zach, Nora Schwegel, Klemens Ablasser, Ewald Kolesnik, Johannes Gollmer, Gert Waltl, Peter P. Rainer, Sarah Verheyen, Andreas Zirlik, Nicolas Verheyen

**Affiliations:** 1Division of Cardiology, Department of Internal Medicine, Medical University of Graz, 8036 Graz, Austria; viktoria.hoeller@medunigraz.at (V.H.); david.zach@medunigraz.at (D.Z.); nora.schwegel@medunigraz.at (N.S.); klemens.ablasser@medunigraz.at (K.A.); ewald.kolesnik@medunigraz.at (E.K.); johannes.gollmer@medunigraz.at (J.G.); peter.rainer@medunigraz.at (P.P.R.); andreas.zirlik@medunigraz.at (A.Z.); 2Institute of Human Genetics, Diagnostic and Research Center for Molecular BioMedicine, Medical University of Graz, 8010 Graz, Austria; heidelis.seebacher@medunigraz.at (H.S.); sarah.verheyen@medunigraz.at (S.V.); 3Division of Cardiology, Department of Internal Medicine, State Hospital (LKH) Graz II, 8020 Graz, Austria; gert.waltl@kages.at

**Keywords:** hypertrophic cardiomyopathy, *MYBPC3*, *MYH7*, genotype-phenotype, speckle tracking, myocardial deformation analysis, longitudinal strain, echocardiography

## Abstract

Accumulating evidence suggests that individuals with sarcomeric hypertrophic cardiomyopathy (HCM) carrying *MYH7* mutations may have a worse prognosis than *MYBPC3* mutation carriers. Myocardial deformation analysis is superior to standard echocardiography in detecting subtle myocardial dysfunction and scar formation, but studies evaluating the association with HCM genotype are scarce. We therefore aimed to compare myocardial strain parameters between *MYBPC3* and *MYH7* mutation carriers with proven HCM. Participants of the prospective Graz HCM Registry carrying at least one causative mutation in *MYBPC3* (*n* = 39) or *MYH7* (*n* = 18) were enrolled. *MYBPC3* mutation carriers were older, predominantly male and more often treated with an implantable cardioverter-defibrillator (39% vs. 0%; *p* = 0.002). Using analyses of covariance, there were no significant differences between *MYBPC3* and *MYH7* mutation carriers with regard to left ventricular global longitudinal strain (estimated marginal means ± standard deviation: −16.9 ± 0.6% vs. −17.3 ± 0.9%; *p* = 0.807) and right ventricular 6-segments endocardial strain (−24.3 ± 1.0% vs. 26.3 ± 1.5%; *p* = 0.285). Our study suggests, that myocardial deformation analysis may not be helpful in concluding on the underlying HCM genotype, and vice versa.

## 1. Introduction

Sarcomeric hypertrophic cardiomyopathy (HCM) is a heritable cardiac disease affecting 1 in 200 to 500 people. While left ventricular hypertrophy is considered the hallmark of sarcomeric HCM, also the right ventricle can be affected. The clinical spectrum ranges from normal to severely impaired myocardial function leading to restrictive cardiomyopathy [1]. Pathogenic or likely pathogenic variants in sarcomere protein encoding genes can be identified in 40–60% of adolescents and adults affected by HCM. *MYBPC3* (encoding cardiac myosin binding protein—C, cMyBPC) and *MYH7* (encoding β cardiac myosin heavy chain) account for the majority of cases [2,3,4].

Accumulating evidence suggests that *MYH7* mutations are associated with an earlier onset of symptoms [5,6,7], more pronounced hypertrophy and poorer prognosis when compared to *MYBPC3* [8,9]. Both, *MYH7* and *MYBPC3* mutations affect the thick filament of the sarcomere [8], but via differing pathways. Regarding *MYH7*, more than 95% of known disease causing variants are missense mutations leading to integration of altered myosin into the sarcomere [10]. Most *MYBPC3* mutations lead to diminished levels of cMyBPC in the sarcomere resulting in haploinsufficiency [11]. Mutations in both genes cause hyperdynamic contraction and poor relaxation of the myocardium [11]. Previous studies failed in the majority to demonstrate phenotypic differences between *MYBPC3* and *MYH7* mutation carriers.

Some imaging studies were suggestive of a more severe phenotype in individuals carrying a pathogenic genetic variant in *MYH7* mutations compared to *MYBPC3*, although differences were often marginal and non-significant [3,8,12]. Other studies found no significant differences with regard to parameters of left ventricular (LV) function and structure, and myocardial scar formation [13,14]. Most studies were, however, limited by their lack to assess myocardial deformation analysis which is superior to standard echocardiography in detecting myocardial scar formation [15]. Moreover, no study has yet reported on association between genotype and right ventricular deformation in sarcomeric HCM.

Therefore, the aim of the present study was to compare echocardiographic characteristics applying myocardial deformation analysis of both the left and the right ventricle, on top of standard echocardiographic parameters in *MYH7-* and *MYBPC3-*associated HCM.

## 2. Materials and Methods

### 2.1. Design and Study Population

This is a cross-sectional analysis of the Graz HCM Registry (EC-No 30-286 ex 17/18). The Graz HCM Registry is a prospective cohort study launched in February 2019 and includes all patients aged over 18 years who are admitted to the HCM outpatient clinic of the Department of Cardiology of the Medical University of Graz, and provide written informed consent for participation. Registry procedures include a systematic transthoracic echocardiographic examination (TTE), a 12-lead electrocardiogram (ECG) and a structured patient interview. 

For the present study we included participants with genetically proven sarcomeric HCM [2]. Patients were excluded if no echocardiographic study recorded within 6 months before or after registry inclusion was judged adequate of LV speckle tracking analysis. Patients were enrolled between February 2019 and June 2021. Clinical characteristics were assessed at the day of echocardiographic examination. 12-lead-ECG, laboratory analysis, and medical history were in most cases assessed on the day of TTE, but at least during a period of 6 months before or after the TTE. 

### 2.2. Clinical Characteristics and Medical History

Clinical and medication history were systematically assessed during the patient interview at registry inclusion and complemented using medical records. NYHA class > II was defined as breathlessness, fatigue or palpitations at less than ordinary activities, e.g., at walking distance less than 1000 m or walking uphill [16]. Sarcomeric HCM was defined as proven mutation either in *MYBPC3* or *MYH7* classified as pathogenic or likely pathogenic and enddiastolic left ventricular wall thickness of ≥13 mm evidenced by TTE [2,17]. Patients carrying mutations in other sarcomere genes were omitted. Left ventricular outflow tract (LVOT) obstruction was defined as maximal left ventricular outflow tract gradient ≥ 30 mmHg either at rest or during provocations such as Valsalva maneuver or bicycle stress testing [18]. Septal reduction therapy was defined as either surgical myectomy or percutaneous transluminal septal myocardial ablation (PTSMA).

### 2.3. Genetic Analysis

The results of performed genetic analyses were collected in the Graz HCM Registry and were retrospectively analyzed. Genetic testing had been performed with DNA from peripheral blood for routine clinical care at different diagnostic laboratories. Panel analyses were used to identify the causal variants in the index patients. Sanger sequencing was used for segregation analysis in family members. Only patients with a confirmed pathogenic or likely pathogenic mutation in *MYBPC3* and *MYH7* were included in the study (Appendix A). Variant classification followed international guidelines for the interpretation of sequence variants [19].

### 2.4. Echocardiographic Assessment and Variables

All patients were examined at rest using Siemens Acuson SC 2000 and a 4Z1 transducer (Siemens AG, Erlangen, Germany). An ECG was recorded during each study to define end-diastole (ED) and end-systole (ES). Images and cine-loops with frame rates from 40 to 80 Hz were stored and digitally archived in IntelliSpace Cardiovascular (ISCV, Philips, Eindhoven, The Netherlands) [20]. All echocardiography derived parameters used in the present study are listed in Table 1. Standard echocardiographic and Doppler measurements were assessed according to EACVI recommendations [21,22,23]. Maximum ED interventricular septum thickness (IVSEDd) was measured in the apical four-chamber view at basal, midventricular and apical levels, respectively.

2D speckle-tracking echocardiography (2D STE) was performed by experienced investigators (D.Z., N.S.) who were blinded to patients’ clinical characteristics, using the vendor-independent post-processing software TomTec-Arena including 2D Cardiac Performance Analysis (TomTec Imaging Systems, Munich, Germany). Cine-loops with the best image quality were selected for 2D STE. In patients with atrial fibrillation, special care was taken to choose cardiac cycles of similar duration. If tracking did not match the visual impression of wall motion, contours were readjusted until optimal tracking was achieved [24].

For the present study, we focused on endocardial strain analysis. LV global longitudinal strain (LV GLS) was calculated using the entire endocardial line length while computing LV deformation from the apical four-, three- and two-chamber views; conducted in two separate cardiac cycles whenever possible and reported as mean values. The endocardium was tracked as the region of interest using two different methods, once automatically generated (i.e., LV GLS auto) and once user-defined (i.e., LV GLS 2DCPA) [17]. Additionally, endomyocardial contouring by 2D Cardiac Performance Analysis was used to calculate ES and ED volumes of each apical view to subsequently compute a triplane ejection fraction of the left ventricle (LVEF) and to determine a triplane left ventricular basal ED diameter (LVEDd basal).

Right ventricular (RV) longitudinal strain was calculated by averaging peak longitudinal systolic strain values of equal segment lengths of the free wall and septum (six segments, i.e., RV4CLS) as well as the free wall only (three segments, i.e., RVFWSL); obtained in the apical four-chamber view. ED was defined by tricuspid valve closure and ES as the moment at which the RV was at its smallest [25,26]. If feasible, measurements were performed in three different cardiac cycles and reported as mean values.

### 2.5. Statistical Analysis

Categorical variables were expressed as counts (percentages), continuous variables were shown as mean ± standard deviation (SD) in case of normal distribution or as medians with interquartile range if non-normally distributed. Distribution of variables was evaluated by test of Kolmogorov-Smirnov, kurtosis, skewness, concordance between the mean and median, and visual inspection. For univariate group comparisons, we used Mann-Whitney U test, Student’s *t*-test or Chi-Square test, as appropriate.

In order to compare echocardiography derived parameters between *MYBPC3* and *MYH7* mutation carriers, analyses of covariance (ANCOVA) were used and adjusted for potentially confounding parameters including age, sex and history of septal reduction therapy. Means are reported as estimated marginal means and standard error derived from multivariate ANCOVA.

Homogeneity of regression slopes was not violated with regard to the dependent variable, as the interaction terms were not statistically significant (*p* > 0.05). The residuals were normally distributed, as determined by the Kolmogorov-Smirnov test (*p* > 0.05). The assumptions of homogeneity of variances were found to be satisfied, as assessed by Levene’s test (*p* > 0.05).

For all statistical analysis IBM SPSS Statistics Version 26 was used. The significance level α was set at 5%. Due to the exploratory character of the study we did not adjust for multiple testing.

## 3. Results

The cohort comprised 57 patients, including 39 (68%) with *MYBPC3* mutation and 18 (32%) with *MYH7* mutation. There was one patient with two disease causing mutations in *MYBPC3* and one patient with a disease-causing mutation in *MYBPC3* and *MYH6.* In 10 patients with *MYBPC3* mutation and in six patients with *MYH7* mutation, there was at least one additional variant of unknown significance (in *MYH6, MYLK2, TTN, FLNC, RYR2, MYO6, TNNT2, MYPN, ILK, COX15, VCL,* respectively).

### 3.1. Clinical Characteristics

Mean age was 49.1 ± 15.2 and 28 patients (49.1%) were females. Patients with *MYBPC3* mutation were significantly older than those with *MYH7* mutation (51.8 ± 14.4 vs. 41.4 ± 14.0 years; *p* = 0.013) and were less often female (*n* = 16, 41% vs. *n* = 12, 67%; *p* = 0.072). Patients carrying a *MYBPC3* mutation had a lower estimated glomerular filtration rate (eGFR; 80.6 ± 22.4 vs. 100.6 ± 31.6 mL/min/1.73 m; *p* = 0.008) and were more often treated with a loop diuretic (21% vs. 0%; *p* = 0.038). Fifteen *MYBPC3* mutation carriers had a previous implantation of an implantable cardioverter-defibrillator (ICD) compared with 0 patients carrying a *MYH7* mutation (39% vs. 0%; *p* = 0.002). Nine patients received an ICD for primary prevention, and six patients for secondary prevention after survived sudden cardiac death (SCD). One SCD survivor with *MYH7* mutation had refused ICD implantation for personal reasons. Atrial fibrillation was more common in *MYBPC3* mutation carriers (31% vs. 6%; *p* = 0.035). Clinical characteristics are listed in more detail in Table 2.

### 3.2. Echocardiographic Parameters

Using analyses of covariates (ANCOVA), there were no significant differences between *MYBPC3* and *MYH7* mutation carriers with regard to LV GLS (estimated marginal means ± standard error: −16.9 ± 0.6 vs.−17.3 ± 0.9; *p* = 0.807), as illustrated in Figure 1. The LV ejection fraction (Simpson method) was slightly lower in *MYBPC3* mutation carriers when compared to *MYH7* mutation carriers without reaching statistical significance (53.03 ± 1.2 vs. 55.4 ± 1.8; *p* = 0.338). Parameters of LV structure were similar between the groups.

The mitral annular early diastolic velocity (e′ average) was significantly higher in the *MYBPC3* group (8.5 ± 0.4 vs. 6.8 ± 0.6 cm/s; *p* = 0.026), but in relation to the transmitral early diastolic velocity (E/e′) no significant difference could be observed (*p* = 0.630). The mean left atrial volume index (LAVi) was similar between both groups (*MYBPC3* 54.3 ± 4.7, *MYH7* 50.1 ± 7.2; *p* = 0.637), as shown in Figure 2.

RV 6-segments endocardial strain was similar between *MYBPC3* and *MYH7* mutation carriers (−24.3 ± 1.0 vs. 26.3 ± 1.5; *p* = 0.285). Fractional area change of the right ventricle (RVFAC) showed significantly lower values in the *MYBPC3* cohort (43.7 ± 1.7 vs. 52.4 ± 2.5; *p* = 0.007). Tricuspid annular plane systolic excursion (TAPSE) showed similar values in both groups (see Figure 3). All findings are resumed in Table 3.

## 4. Discussion

This is the first study describing detailed genotype-phenotype correlations in sarcomeric HCM focusing on myocardial deformation markers of both RV and LV. There were no significant differences between *MYBPC3* and *MYH7* mutation carriers with regard to LV and RV longitudinal strain, respectively. In multivariate statistical models, only RVFAC and e’ were significantly different. However, while *MYBPC3* mutation carriers had lower RVFAC suggesting a poorer RV function, they presented with higher e’ indicating better LV relaxation. In light of this ambiguity, these findings may be interpreted as effects of chance rather than as results of phenotypic differences related to the underlying mutations. The higher rate of ICD implantations in the MYBPC3 group may have confounded these associations as well.

The reason for the suggestively earlier onset of symptoms, worse prognosis and more pronounced hypertrophy in *MYH7* mutations [5,6,7,8,9] could be explained by functional differences between the proteins coded by *MYBPC3* and *MYH7*. Cycling interaction between actin and myosin drives sarcomeric contraction through sliding of thick and thin filaments past one another, creating force that allows cardiomyocytes to contract and relax. Cardiac MyBPC (cMyBPC) regulates myocardial contractility, with reduced cMyBPC levels leading to hypercontractility and impaired relaxation. *MYBPC3* mutations are supposed to cause HCM by haploinsufficiency. Myosin is a mechanoenzyme that drives ventricular contraction and produces force when binding actin and hydrolyzing ATP [4]. Most HCM causing *MYH7* mutations cluster between residues 181 and 937, forming the myosin head domain and approximately 20% are located in the coiled coil region forming the thick filament [27]. Mutant myosins show altered parameters of myocardial contraction like actin gliding velocity, intrinsic force production, cross—bridge cycling kinetics, calcium sensitivity of force generation and acto-myosin ATPase activity leading to hypercontractility and impaired relaxation [27,28,29,30]. Consistent with previous results [31], causal *MYH7* variants in our cohort were predominantly missense mutations (see Appendix A).

Pathogenic variants in *MYBPC3* are predominantly truncating [7,32,33], which is in accordance with our results. Causal *MYBPC3* variants in our cohort were primarily frameshift and splice site mutations (see Appendix A). Only a few probands had disease causing missense mutations and there was one proband with an in-frame deletion and one with a nonsense mutation, respectively (see Appendix A). *MYBPC3-*associated HCM shows a later disease onset with a variability in the rate of progression even within a family, influenced by lifestyle, environment and other genetic factors [5,34]. The later age of onset is discussed as explanation for a relatively high proportion of founder mutations. All of them result in a shortened cMyBPC. Members of two families of our cohort carry a Dutch founder mutation (c.2864_2865delCT) [33]. A Tuscany founder mutation (c.772G > A) was detected in members of five families of our cohort [35].

Our results showed a higher age of disease onset in *MYBPC3* patients. This is in line with a recent meta-analysis including 51 studies with 7675 HCM patients, where mutations in *MYH7* were associated with earlier age of onset and higher risk of sudden cardiac death when compared to *MYBPC3* [9]. In contrast, an earlier meta–analysis comprising 18 studies with 2459 patients found no differences in terms of symptoms, age of onset and grade of left ventricular hypertrophy between *MYBPC3* and *MYH7* mutation carriers [36]. 

Interestingly, *MYBPC3* mutations carriers had significantly more often a history of ICD implantation. This is not in line with other and larger studies attributing a higher risk of SCD to *MYH7* [8,9]. Patients with missense mutations affecting the actin binding site or the head–rod portion of β MHC showed decreased survival [37]. For instance, the p.Arg453Cys mutation in *MYH7* is associated with a high incidence of end-stage heart failure and premature death [38]. Similarly, a significantly higher proportion of the *MYBPC3* group had history of atrial fibrillation which is not in line with previous studies [9,39,40]. Both of these controversial observations are likely a consequence of referral bias in a tertiary HCM care center which is a well-known phenomenon in epidemiological research on HCM [41].

Only a few studies compared echocardiographic parameters between HCM genotypes although none explicitly reported results of RV myocardial deformation analyses. One recent multicenter study including 63 adult individuals evaluated the association between mutations in both genes and phenotypes in patients with sarcomeric HCM. They found that patients carrying a *MYH7* mutation were similar to *MYBPC3* carriers in the majority of measured echocardiographic parameters. Only systolic anterior motion of the mitral valve and mitral valve calcification were significantly more common in *MYH7* mutation carriers [8]. It is well accepted that analysis of myocardial deformation markers is superior to standard echocardiographic parameters in detecting scar formation [15], predicting arrhythmias [42] and cardiovascular outcomes in HCM [43]. Evidence on differences of myocardial deformation markers between distinct HCM susceptibility genes is scarce. One previous study found that LV strain is similar between *MYBPC3* and *MYH7* mutation carriers. In this study the authors concluded that LV morphology rather than genotype predicts myocardial deformation markers in HCM [44]. However, RV myocardial deformation analyses were not included in their report.

Particular strengths of our study include its novelty, since myocardial deformation, particularly of the RV, has not been sufficiently investigated in patients with sarcomeric HCM. A further strength of our study is the high quality of strain measurements which were performed by blinded investigators analyzing several cine loops per patient. 

Limitations of our study include the relatively low sample size and the single-center design. Moreover, characteristics of our cohort may be confounded by referral bias which may be inherent to our tertiary care setting, although we performed multivariate analysis to minimize this bias. Nevertheless, results may not be generalizable to other HCM populations.

## 5. Conclusions

Echocardiographic myocardial deformation parameters of both RV and LV were similar between *MYBPC3* and *MYH7* mutation carrying individuals with sarcomeric HCM. Myocardial deformation analysis may not be helpful in concluding on the underlying HCM genotype, and vice versa.

## Figures and Tables

**Figure 1 genes-12-01469-f001:**
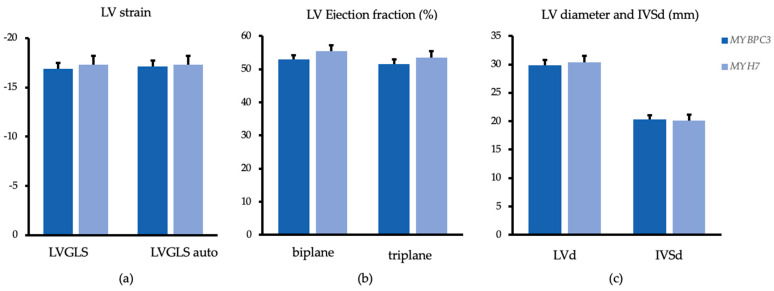
Echocardiographic analysis of the left ventricle. Estimated marginal means and standard error adjusted for age, sex and septum reduction therapy, compared between *MYBPC3* (dark blue) and *MYH7* (light blue) mutation carriers. (**a**) Left ventricular strain analysis user—(LVGLS) (*p* = 0.807) and automatically (LVGLS auto) (*p* = 0.892) generated; (**b**) Ejection fraction of the left ventricle (%) measured biplane (Simpson) (*p* = 0.338) and triplane (*p* = 0.410) in %; (**c**) enddiastolic basal diameter of the left ventricle (LVd) (*p* = 0.693) and maximal enddiastolic thickness of the interventricular septum (IVSd) (*p* = 0.897).

**Figure 2 genes-12-01469-f002:**
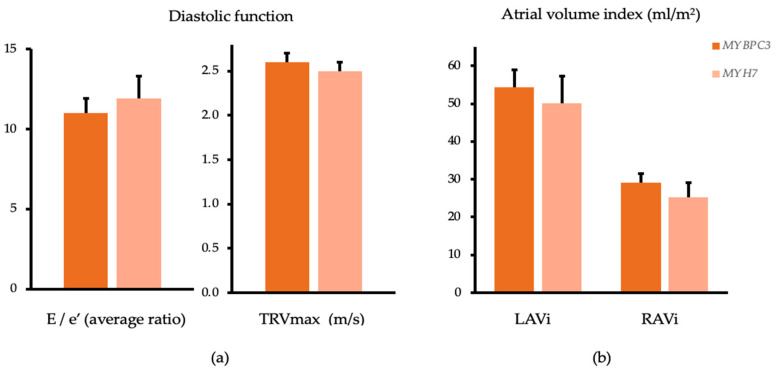
Echocardiographic analysis of the diastolic function and the atrial volume. Estimated marginal means and standard error adjusted for age, sex and septum reduction therapy compared between *MYBPC3* (dark orange) and *MYH7* (light orange) mutation carriers. (**a**) ratio of early transmitral velocity to average velocity of the transmitral annulus (E/e′) (*p* = 0.630) and maximal tricuspid regurgitation velocity (TRVmax) (*p* = 0.390); (**b**) left atrial volume index (LAVi) (*p* = 0.637) and right atrial volume index (RAVi) (*p* = 0.443).

**Figure 3 genes-12-01469-f003:**
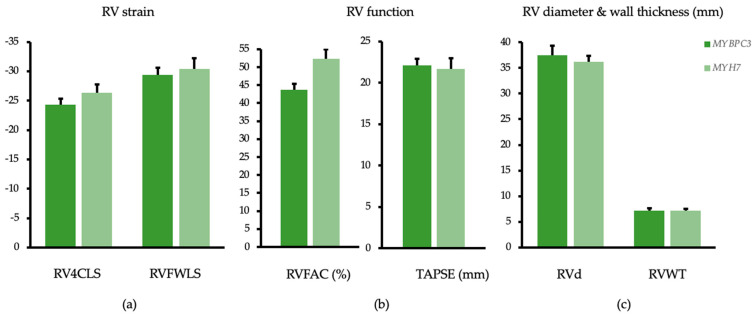
Echocardiographic analysis of the right ventricle. Estimated marginal means and standard error adjusted for age, sex and septum reduction therapy and compared between *MYBPC3* (dark green) and *MYH7* (light green) mutation carriers. (**a**) Right ventricular strain analysis (RV4CLS) (*p* = 0.285) and strain analysis of the right free wall (RVFWLS) (*p* = 0.643); (**b**) right ventricular fractional area change (RVFAC) (*p* = 0.007), tricuspid annular plane systolic excursion (TAPSE) (*p* = 0.798); (**c**) enddiastolic basal diameter of the right ventricle (RVd) (*p* = 0.551) and enddiastolic thickness of the right ventricular free wall (RVWT) (*p* = 0.917).

**Table 1 genes-12-01469-t001:** Echocardiography derived parameters.

Chamber	Parameter
Left ventricle	LVEF, Simpson’s biplane (%) LVEF, triplane, 2dCPA (%)
	Transmitral *E* velocity (cm/s)
	Septal annular *e*′ velocity (cm/s)
	Lateral annular *e*′ velocity (cm/s)
	LV *E*/*e*′ (average) ratio LV GLS, auto-strain (%) LV GLS, 2dCPA (%)
Left atrium	LAVi (mL/m^2^)
Right ventricle	RV basal ED diameter (mm)
	RV wall thickness (mm)
	TAPSE (mm) TRVmax (m/s)
	Fractional area change (%) RVLS 6 segments, 2dCPA (%) RVLS free wall, 2dCPA (%)
Right atrium	RAVi (mL/m^2^)

Abbreviations: LV, left ventricular; EF, ejection fraction; 2dCPA, two-dimensional cardiac performance analysis; E, transmitral early diastolic velocity; *e′*, mitral annular early diastolic velocity; LAVi, left atrial volume index; GLS, global longitudinal strain; RV, right ventricular; ED, end-diastolic; TAPSE, tricuspid annular plane systolic excursion; TRVmax, maximal tricuspid regurgitation velocity; RLVS, right ventricular longitudinal strain; RAVi, right atrial volume index.

**Table 2 genes-12-01469-t002:** Clinical characteristics and medical history.

	Whole Cohort (*n* = 57)	*MYBPC3* (*n* = 39)	*MYH7* (*n* = 18)	*t*-Test/Pearson Chi^2^-Test
Characteristics	*n* (%) or Mean ± SD or Median (IQR)	*n* (%) or Mean ± SD or Median (IQR)	*n* (%) or Mean ± SD or Median (IQR)	*p* Value
Female	28 (49.1)	16 (41.0)	12 (66.7)	*p* = 0.072
Age (years)	49.07 ± 15.24	51.82 ± 14.40	41.44 ± 13.95	***p* = 0.013**
Age at first diagnosis (years) ^1^	40.78 ± 15.55	41.80 ± 14.61	36.4 ±16.91	*p* = 0.259
BMI (kg/m^2^) ^2^	27.52 ± 5.90	26.55 ± 4.47	29.59 ± 7.91	*p* = 0.143
RR systolic (mmHg) ^2^	134.73 ± 18.31	135.88 ± 19.13	132.33 ± 16.71	*p* = 0.505
RR diastolic (mmHg) ^2^	80.32 ± 10.04	79.71 ± 10.18	81.61 ± 9.91	*p* = 0.513
Heartrate (beats/min)	67.39 ± 10.20	68.44 ± 10.04	65.11 ± 10.44	*p* = 0.256
NYHA > II	19 (33.3)	15 (38.5)	4 (22.3)	*p* = 0.227
LVOT obstruction	22 (38.6)	16 (41.0)	6 (33.3)	*p* = 0.579
Loop diuretics	8 (14.0)	8 (20.5)	0	***p* = 0.038**
MRA	5 (8.8)	4 (10.3)	1 (5.6)	*p* = 0.560
Verapamil	5 (8.8)	4 (10.3)	1 (5.6)	*p* = 0.560
Amiodarone	3 (5.3)	2 (5.1)	1 (5.6)	*p* = 0.946
ACEI/ARB/ARNI	12 (21.2)	11 (28.2)	1 (5.6)	*p* = 0.051
NTproBNP (pg/mL)	536 (193, 1470)	817 (197, 2802)	390 (164, 789)	*p* = 0.158
eGFR (ml/min/1.73 m)	86.89 ± 27.07	80.55 ± 22.4	100.62 ± 31.62	***p* = 0.008**
**History**				
Surgical myectomy	3 (5.3)	1 (2.6)	2 (11.1)	*p* = 0.179
PTSMA	9 (15.8)	7 (17.9)	2 (11.1)	*p* = 0.510
Mitral valve replacement/repair	0	0	0	
Survived sudden cardiac death	7 (12.3)	6 (15.4)	1 (5.6) *	*p* = 0.293
Hospitalization for WHF	8 (14.0)	7 (17.9)	1 (5.6)	*p* = 0.211
Arterial hypertension	23 (40.4)	17 (43.6)	6 (33.3)	*p* = 0.463
Stroke	4 (7.0)	4 (10.3)	0	*p* = 0.132
PE	3 (5.3)	2 (5.1)	1 (5.6)	*p* = 0.946
AF	13 (22.8)	12 (30.8)	1 (5.6)	***p* = 0.035**
permanent	4 (30.8 **)	4 (33.3 **)	0	*p* = 0.159
paroxysmal	8 (61.5 **)	8 (66.7 **)	1 (100 **)	*p* = 0.150
**ECG**				
Sinus rhythm	48 (84.3)	30 (77)	18 (100)	***p* = 0.026**
Ventricular stimulated rhythm	5 (8.8)	5 (12.8)	0	*p* = 0.112
Atrial fibrillation	4 (7.0)	4 (10.3)	0	*p* = 0.159
QRS duration (ms)	111.43 ± 33.67	113.8 5 ± 37.72	104.67 ± 21.65	*p* = 0.251
LBBB	19 (33.3)	11 (28.2)	8 (44.4)	*p* = 0.227
RBBB	11 (19.3)	8 (20.5)	3 (16.7)	*p* = 0.732
**Devices**				
Pacemaker	15 (26.3)	15 (38.5)	0	***p* = 0.002**
CRT	1 (1.8)	1 (2.6)	0	*p* = 0.493
ICD	15 (26.3)	15 (38.5)	0 *	***p* = 0.002**
ICD shock	4 (26.7 **)	4 (26.7 **)		
Primary prevention	9 (60.0 **)	9 (60.0 **)		
Secondary prevention	6 (40.0 **)	6 (40.0 **)		

^1^ *MYBPC3* (*n* = 35) *MYH7* (*n* = 15). ^2^ *MYBPC3* (*n* = 38). * patient refused ICD therapy. ** Relative percentage. Bold, statistically significant. Abbreviations: SD, standard deviation, IQR, interquartile range; BMI, body mass index; RR, blood pressure; NYHA, New York Heart Association; LVOT, left ventricular outflow tract; MRA, mineral corticoid receptor antagonist; ACEI, angiotensin converting enzyme inhibitor; ARB, angiotensin receptor blocker; ARNI, angiotensin receptor neprilyin inhibitor; NTproBNP, N-terminal pro-brain natriuretic peptide; eGFR, estimated glomerular filtration rate; PTSMA, percutaneous transluminal septal myocardial ablation; WHF, worsening heart failure; PE, pulmonary embolism; AF, atrial fibrillation; LBBB, left bundle branch block; RBBB, right bundle branch block; CRT, cardiac resynchronization therapy; ICD, implantable cardioverter-defibrillator.

**Table 3 genes-12-01469-t003:** Echocardiographic parameters compared between *MYBPC3* and *MYH7* mutation carriers.

	*MYBPC3* (*n* = 39)		*MYH7* (*n* = 18)		*t*-Test	*ANCOVA*
	Not Adjusted	Adjusted	Not Adjusted	Adjusted		
	Mean ± SD or Median (IQR)	Mean ± SE	Mean ± SD or Median (IQR)	Mean ± SE	*p* Value	*p* Value
Sinus rhythm during TTE *n* (%)	30 (77)		18 (100)			
heart rate during TTE (beats/min)	68.44 ± 10.04	68.9 ± 1.7	65.11 ± 10.44	64.0 ± 2.6	*p* = 0.256	*p* = 0.139
LV average loops	2 ± 0		2 ± 0			
RV average loops	2.53 ± 0.6		2.47 ± 0.7			
LV						
LVGLS auto ^1^	−16.7 ± 4.1	−17.2 ± 0.6	−18.3 ± 2.6	−17.3 ± 0.9	*p* = 0.138	*p* = 0.892
LVGLS 2dCPA	−16.5 ± 4.0	−16.9 ± 0.6	−18.3 ± 2.8	−17.3 ± 0.9	*p* = 0.088	*p* = 0.807
LVEF Simpson biplane (%) ^2^	52.9 ± 8.0	53.03 ± 1.2	56.3 ± 4.8	55.4 ± 1.8	*p* = 0.100	*p* = 0.338
LVEF triplane (%)	51.1 ± 8.6	51.6 ± 1.3	54.7 ± 4.8	53.5 ± 1.9	*p* = 0.105	*p* = 0.410
E/e′ average (ratio) ^3^	11.4 ± 5.8	11.0 ± 0.9	11.1 ± 4.6	11.9 ± 1.4	*p* = 0.652	*p* = 0.630
E (cm/s) ^3^	83.0 ± 27.1	84.3 ± 4.6	79.4 ± 22.6	76.7 ± 7.1	*p* = 0.647	*p* = 0.397
e′ average (cm/s) ^3^	8.1 ± 2.7	8.5 ± 0.4	7.7 ± 2.4	6.8 ± 0.6	*p* = 0.892	***p* = 0.026**
IVSd maximal (mm)	20.8 ± 4.6	20.3 ± 0.8	19.2 ± 4.8	20.1 ± 1.1	*p* = 0.226	*p* = 0.897
IVSd basal (mm)	15.2 ± 3.8	14.9 ± 0.6	13.5 ± 3.8	14.3 ± 0.9	*p* = 0.125	*p* = 0.626
IVSd midventricular (mm)	20.7 ± 4.6	20.2 ± 0.8	19.1 ± 4.7	20.1 ± 1.1	*p* = 0.233	*p* = 0.911
IVSd apical (mm)	14.4 ± 4.6	14.1 ± 0.7	13.7 ± 4.3	14.3 ± 1.1	*p* = 0.593	*p* = 0.868
LVEDd (cm)	3.03 ± 0.54	2.98 ± 0.1	2.94 ± 0.36	3.04 ± 0.11	*p* = 0.527	*p* = 0.693
LAVi MOD (mL/m^2^) ^4^	50.2 (37.8, 68.8)	54.3 ± 4.7	42.5 (33.5, 49.9)	50.1 ± 7.2	*p* = 0.097	*p* = 0.637
RV						
RVLS 6 segments 2dCPA ^5^	−24.1 ± 6.3	−24.3 ± 1.0	−26.9 ± 3.8	−26.3 ± 1.5	*p* = 0.062	*p* = 0.285
RVLS free wall 2dCPA ^5^	−29.1 ± 7.4	−29.4 ± 1.2	−30.9 ± 4.2	−30.4 ± 1.8	*p* = 0.395	*p* = 0.643
RVFAC (%) ^2^	43.5 ± 10.9	43.7 ± 1.7	52.8 ± 7.4	52.4 ± 2.5	***p* = 0.002**	***p* = 0.007**
TRVmax (m/s) ^6^	2.6 ± 0.4	2.6 ± 0.1	2.5 ± 0.2	2.5 ± 0.1	*p* = 0.206	*p* = 0.390
TAPSE (mm) ^7^	21.9 ± 4.5	22.1 ± 0.8	22.2 ± 5.0	21.7 ± 1.3	*p* = 0.823	*p* = 0.798
RVEDd basal (mm) ^2^	38.4 ± 7.1	37.5 ± 1.2	34.3 ± 8.2	36.2 ± 1.8	*p* = 0.059	*p* = 0.551
RVWT (mm) ^8^	7.2 ± 1.7	7.2 ± 0.3	7.2 ± 2.0	7.2 ± 0.5	*p* = 0.950	*p* = 0.917
RA Area (cm^2^) ^9^	18.5 ± 1.0	18.3 ± 1.0	16.6 ± 1.6	17.3 ± 1.6	*p* = 0.045	*p* = 0.619
RAVi (mL/m^2^) ^2,9^	31.4 ± 19.1	29.1 ± 2.5	20.3 ± 7.4	25.3 ± 3.9	***p* = 0.003**	*p* = 0.443

^1^ *MYBPC3* (*n* = 34), *MYH7* (*n* = 17). ^2^
*MYBPC3* (*n* = 38). ^3^
*MYBPC3* (*n* = 33), *MYH7* (*n* = 15). ^4^
*MYBPC3* (*n* = 35) *MYH7* (*n* = 16). ^5^ *MYBPC3* (*n* = 32), *MYH7* (*n* = 15). ^6^
*MYBPC3* (*n* = 25), *MYH7* (*n* = 7). ^7^ *MYBPC3* (*n* = 35), *MYH7* (*n* = 14). ^8^
*MYBPC3* (*n* = 31), *MYH7* (*n* = 13). ^9^
*MYH7* (*n* = 17). Bold, statistically significant. Abbreviatons: SD, standard deviation; IQR, interquartile range; SE, standard error; ANCOVA, analysis of covariance; TTE, transthoracic echocardiography; LV, left ventricule/ventricular; RV, right ventricle/ventricular; GLS, global longitudinal strain; auto, automatically generated; 2dCPA, 2D cardiac performance analysis; EF, ejection fraction; *E*, transmitral early diastolic velocity; *e′*, mitral annular early diastolic velocity; IVS, interventricular septum; d, diameter; ED, end-diastolic; LAVi, left atrial volume index; MOD, method of discs; LS, longitudinal strain; FAC, fractional area change; TRVmax, maximal tricuspid regurgitation velocity; TAPSE, tricuspid annular plane systolic excursion; WT, wall thickness; RAVi, right atrial volume index.

## Data Availability

The data presented in this study are available on request from the corresponding author.

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
