# Peer review of "Myocardial Deformation Analysis in MYBPC3 and MYH7 Related Sarcomeric Hypertrophic Cardiomyopathy—The Graz Hypertrophic Cardiomyopathy Registry"

_genes, 2021, doi:10.3390/genes12101469_

Round 1
Reviewer 1 Report
This is an interesting study to explore the association of MYH7 and MYBPC3 mutation in sarcomeric HCM. Although the current study did not detect significant differences between these two mutations with HCM, it is not clear if these results could be conclusive at this point. The authors indeed discussed the limitations, specifically low sample size and single-center data collection. However, it would also be worth noting the observation may be affected by a relatively higher rate of patients in MYBPC3 mutations with device implantation. Overall, the potential importance of this study for future continued investigation overweighed the limitations and inconclusive results at this stage.
Author Response
Response to the reviewers
We thank the Editor and the reviewers for this opportunity to submit a revised version of our Original Manuscript (ID genes-1368024). We have invested all efforts to adapt the manuscript according to the reviewers’ comments. The revised parts have been marked in yellow. We also provide a point-by-point reply to the comments. We feel that our manuscript has improved substantially due to the editor’s and reviewers’ comments and are confident that it should now meet the high standards of Genes.
Reviewer #1:
This is an interesting study to explore the association of MYH7 and MYBPC3 mutation in sarcomeric HCM. Although the current study did not detect significant differences between these two mutations with HCM, it is not clear if these results could be conclusive at this point. The authors indeed discussed the limitations, specifically low sample size and single-center data collection. However, it would also be worth noting the observation may be affected by a relatively higher rate of patients in MYBPC3 mutations with device implantation.
Response:
We thank the reviewer for this important comment. Indeed, our study is limited by its low sample size and the single-center character as already outlined in the limitations section. We agree that the higher rate of ICD implantations in the MYBPC3 group may confound the few between group differences and have therefore added the following sentence in the Discussion Section:
Line 247: “Also the higher rate of ICD implantations in the MYBPC3 group may have confounded these associations.”
The main focus of our work was to present data on myocardial deformation imaging. In order to strengthen this focus, we have deleted the last sentence of the Abstract “The association between genotype and the risk of arrhythmias warrants further studies.” and also the last sentence of the Conclusion “The association between genotype and the risk of arrhythmias warrants further studies.”.
Overall, the potential importance of this study for future continued investigation overweighed the limitations and inconclusive results at this stage.
Response:
We thank the reviewer for this overall positive statement.

Reviewer 2 Report
Höller and colleagues presented the original work pointing out that there is not a significant difference among carriers of mutations MYH7 and MYBPC3 using myocardial deformation analysis in patients with sarcomeric hypertrophic cardiomyopathy (HCM). The work did not provide any novel data, it only confirmed the already known fact from the literature, that it is not possible to estimate the underlying mutations according to the myocardial deformation analysis. The authors also pointed out some data that contradict the available literature which might be the result of a "dirty" sample or limited sample size rather than an exception of the Austrian population as authors explained (pointed out the necessity of further investigation).
The authors in this original paper wanted to check whether is possible to evaluate the underlying genotype in patients with sarcomeric hypertrophic cardiomyopathy (HCM) via parameters provided by myocardial deformation analysis. They confirmed that analysis of myocardial deformity is not helpful for this purpose. More precisely, there is not a significant difference among carriers of mutations MYH7 and MYBPC3 using myocardial deformation analysis in patients with HCM according to their data.
These data are not new and only confirm the already available data.
The paper confirms the fact mention in the first answer (above), which is in line with the literature, so nothing novel.
- In addition, the paper contains some data that are contradictory compared to the available literature. However, those data, as I mentioned previously, might be the result of a dirty" sample or limited sample size rather than an exception of the Austrian population as the authors explained.
I would suggest enrolling more patients because a larger sample will avoid bias. Also, it might be interesting to follow this cohort in the future to confirm the correlation with a specific mutation and higher risk of adverse outcomes.
Regarding the main question, I would say yes. They confirmed that is not possible to evaluate the underlying genotype parameters provided by myocardial deformation analysis. However, from the line 279-288, the opposite data might need further investigation and larger cohort. There are very little data in this paper.
Author Response
Response to the reviewers
We thank the Editor and the reviewers for this opportunity to submit a revised version of our Original Manuscript (ID genes-1368024). We have invested all efforts to adapt the manuscript according to the reviewers’ comments. The revised parts have been marked in yellow in the manuscript. We also provide a point-by-point reply to the comments. We feel that our manuscript has improved substantially due to the editor’s and reviewers’ comments and are confident that it should now meet the high standards of Genes.
Reviewer #2:
Höller and colleagues presented the original work pointing out that there is not a significant difference among carriers of mutations MYH7 and MYBPC3 using myocardial deformation analysis in patients with sarcomeric hypertrophic cardiomyopathy (HCM). The work did not provide any novel data, it only confirmed the already known fact from the literature, that it is not possible to estimate the underlying mutations according to the myocardial deformation analysis.
Response: We thank the reviewer for this comment. However, we kindly disagree with the statement that our study does not provide novel data. In fact, this is the first study reporting associations between genotype and right ventricular myocardial deformation analysis in sarcomeric HCM. We have tried to emphasize this throughout the manuscript by adding the following:
Introduction:
“While left ventricular hypertrophy is considered the hallmark of sarcomeric HCM, also the right ventricle can be affected.”
“Moreover, no study has yet reported on association between genotype and right ventricular deformation in sarcomeric HCM.”
Discussion:
“This is the first study describing detailed genotype-phenotype correlations in sarco-meric HCM focusing on myocardial deformation markers of both RV and LV.”
“Only a few studies compared echocardiographic parameters between HCM geno-types although none explicitly reported results of RV myocardial deformation analyses.”
“However, RV myocardial deformation analyses were not included in their report.”
“Particular strengths of our study include its novelty, since myocardial deformation, particularly of the RV, has not been sufficiently investigated in patients with sarcomeric HCM.”
The authors also pointed out some data that contradict the available literature which might be the result of a "dirty" sample or limited sample size rather than an exception of the Austrian population as authors explained (pointed out the necessity of further investigation).
Response:
We thank the reviewer for highlighting these limitations of our work which are, however, inherent to any single center cohort study reporting on rare disease such as sarcomeric HCM.
The present study includes all HCM patients that have been investigated in our tertiary care center since February 2019. We agree that referral bias may have affected the imbalances between both the MYBPC3 and the MYH7 group. However, referral bias is a common phenomenon in epidemiological HCM research as for instance stated by Maron and colleagues (Clinical Course of Hypertrophic Cardiomyopathy in a Regional United States Cohort, JAMA 1999). This reference has been added in the Discussion section. In this context, we consider publication of data from HCM centers highly relevant to the international community because they reflect daily clinical practice.
We have added in the Discussion Section:
“Both these controversial observations are likely a consequence of referral bias in a tertiary HCM care center which is a well-known phenomenon in epidemiological research on HCM [41].”
We agree that our data may not be suited to draw conclusions on Austrian peculiaritites. Therefore we have deleted this point in the Discussion section.
The authors in this original paper wanted to check whether is possible to evaluate the underlying genotype in patients with sarcomeric hypertrophic cardiomyopathy (HCM) via parameters provided by myocardial deformation analysis. They confirmed that analysis of myocardial deformity is not helpful for this purpose. More precisely, there is not a significant difference among carriers of mutations MYH7 and MYBPC3 using myocardial deformation analysis in patients with HCM according to their data.
These data are not new and only confirm the already available data.
The paper confirms the fact mention in the first answer (above), which is in line with the literature, so nothing novel.
Response:
As answered in our first response to this reviewer we have added several sections that underline the novel aspects of our work, particularly the report of RV myocardial deformation parameters.
In addition, the paper contains some data that are contradictory compared to the available literature. However, those data, as I mentioned previously, might be the result of a dirty" sample or limited sample size rather than an exception of the Austrian population as the authors explained.
Response:
We again thank the reviewer for adressing the limitations of our study. We have re-arranged the limitations section:
“Limitations of our study include the relatively low sample size and the single-center design. Moreover, characteristics of our cohort may be confounded by referral bias which may be inherent to our tertiary care setting, alhtough we have tried to compensate for this by performing multivariate analysis. Nevertheless, results may not be generalizable to other HCM populations.”
Moreover, we have attenuated the interpretations of our findings not in line with previous studies by replacing “contradictory” by “not in line with”.
I would suggest enrolling more patients because a larger sample will avoid bias. Also, it might be interesting to follow this cohort in the future to confirm the correlation with a specific mutation and higher risk of adverse outcomes.
Regarding the main question, I would say yes. They confirmed that is not possible to evaluate the underlying genotype parameters provided by myocardial deformation analysis. However, from the line 279-288, the opposite data might need further investigation and larger cohort. There are very little data in this paper.
Response
As stated in our previous responses, this is a single-center study enrolling all patients from our HCM tertiary care center. We have included all patients that have consulted our center since February 2019 and fulfilled our predefined inclusion criteria for the present analysis. If referall bias is the causal confounding factor leading to imbalances between the sarcomere gene groups, we consider it unlikely that enrollment of more patients, for example from before 2019, would attenuate between-group discrepancies because the underlying selection bias would not be altered. Moreover, selection bias is a common phenomenon in HCM research and not specific to our report. In order to compensate for this limitation, we performed multivariate analyses to adjust for these between group differences.
The main focus of our work was to present data on myocardial deformation imaging. In order to strengthen this focus, we have deleted distracting sentences:
Abstract: “The association between genotype and the risk of arrhythmias warrants further studies.”
Conclusion: “The association between genotype and the risk of arrhythmias warrants further studies.”.
We agree that it would be interesting to collect outcomes of our cohort. However, since we have just gathered the baseline characteristics the rate of outcomes will be too low to have sufficient power to perform statistical analyses. Moreover, longitudinal analyses were not predefined endpoints of the present work and are beyond the scope to describe cross-sectional assocations between myocardial deformation analysis and genotype.

Round 2
Reviewer 2 Report
I am satisfied with the author's corrections in the manuscript. To avoid misunderstandings, they emphasized the limitations and corrected certain sentences in the discussion section that I did not approve of in the first version, so I must say that it looks much better now. Regarding the last paragraph of the discussion, although they discussed the "issue of bias" in the comments, they changed it in a manuscript that I considered very reasonable. Furthermore, they added additional explanations that are appropriate in this context.